# Pattern of Expression of Genes Involved in Systemic Inflammation and Glutathione Metabolism Reveals Exacerbation of COPD

**DOI:** 10.3390/antiox13080953

**Published:** 2024-08-06

**Authors:** Ingrid Oit-Wiscombe, László Virág, Kalle Kilk, Ursel Soomets, Alan Altraja

**Affiliations:** 1Department of Pulmonology, University of Tartu, 50406 Tartu, Estonia; 2Institute of Biomedicine and Translational Medicine, University of Tartu, 50411 Tartu, Estonia; kalle.kilk@ut.ee (K.K.);; 3Centre of Excellence for Genomics and Translational Medicine, University of Tartu, 50411 Tartu, Estonia; 4Department of Medical Chemistry, Faculty of Medicine, University of Debrecen, 4032 Debrecen, Hungary; lvirag@med.unideb.hu; 5HUN-REN-DE Cell Biology and Signaling Research Group, 4032 Debrecen, Hungary; 6Lung Clinic, Tartu University Hospital, 50406 Tartu, Estonia

**Keywords:** COPD, COPD exacerbation, oxidative stress, glutathione, metabolism, systemic inflammation

## Abstract

To test the hypothesis that they serve as systemic biomarkers of chronic obstructive pulmonary disease (COPD), we profiled the mRNA expression of enzymes connected to systemic inflammation and GSH metabolism in peripheral blood mononuclear cells (PBMCs). These were taken from patients displaying acute exacerbation of COPD (AE-COPD) and stable COPD, and also from non-obstructive smokers and non-smokers. The expression of poly(ADP-ribose) polymerase-1 was increased, but that of histone deacetylase 2 was decreased in association with AE-COPD. The expression of modulatory subunit of glutamyl–cysteine ligase was higher and that of its catalytic subunit, together with the expression of dipeptidyl peptidase 4, was lower in COPD patients compared with non-obstructive smokers and non-smokers. Leukotriene A4 hydrolase saw increased expression in patients with COPD according to disease severity compared to non-obstructive individuals, whereas the expression of GSH peroxidase increased in non-obstructive smokers and COPD patients with the growing number of pack-years smoked. The results corroborate COPD and its acute exacerbation as a complex systemic disorder demonstrating distinct associations with the expression of enzymes linked to inflammation and the regulation of GSH metabolism.

## 1. Introduction

Cigarette smoking, the most common risk factor of chronic obstructive pulmonary disease (COPD), prompts inflammation and oxidative stress (OS), both in the airways and systemically [1,2]. Developing wider knowledge of circulating biomarkers is important to facilitating a proper understanding of the pathophysiology of COPD. Acute exacerbations of COPD (AE-COPDs) are common disease-escalating complications [3]. Although, in practice, AE-COPDs are predicted by a history of previous events and deteriorating airflow limitation [3,4], these data lack sufficient precision to predict AE-COPDs and mortality [3].

Pro-inflammatory enzymes 5-lipoxygenase (5-LO), leukotriene A4 hydrolase (LTA_4_H), dipeptidyl peptidase 4 (DPP4), and cyclooxygenase-2 (COX-2), and anti-inflammatory enzymes poly(ADP-ribose) polymerase-1 (PARP-1) and histone deacetylase 2 (HDAC2), are connected to COPD and underlying systemic inflammation [2,5,6,7]. Cyclooxygenases (COXs) and lipoxygenase (LO) produce eicosanoids, which are potent inflammatory mediators, from arachidonic acids [8]. LTA_4_H is a major enzyme of the 5-LO pathway [9]. DPP4, a multifunctional protein, plays a role in the post-translational modification of hormones and chemokines, T-cell activation, cell adhesion, and apoptosis [7,10]. PARP-1 is a post-translational modifying enzyme with a primary role in DNA repair [11]. We have previously shown that PARP-1 activity is increased in peripheral blood mononuclear cells (PBMCs) in patients with COPD [2]. Increased DNA damage and increased PARP activity were shown in PBMCs in correlation with COPD severity, indicating that intensified DNA repair occurs in the PBMCs of patients with higher COPD stages [2].

OS plays a key role in the development and progression of COPD, chiefly due to increased oxidant exposure from cigarette smoke (CS) [12]. Prolonged airway inflammation and increased OS are associated with the increased frequency and severity of AE-COPDs [13]. OS is also connected with glucocorticosteroid resistance [14] and decreased antioxidative potential: the protective antioxidant levels are significantly depleted in the alveolar macrophages of patients with COPD [15]. Glutathione (GSH) is predominantly known as an antioxidant [16]. Therefore, we set the aim of exploring the relationship of systemic inflammation and the enzymes involved in the GSH metabolism pathway [superoxide dismutase 1 (SOD1), GSH synthetase (GSS), the catalytic and modulatory subunits of glutamyl–cysteine ligase (GCLC and GCLM), GSH peroxidase (GPx), and GSH reductase (GSR)] with smoking data, lung function parameters, the presence of exacerbations, and the Global Initiative for COPD (GOLD) classification categories of COPD.

## 2. Materials and Methods

The protocol of this prospective cross-sectional study was approved by the Ethics Committee on Human Research of the University of Tartu (protocol nr: 170/T-22) and all procedures were conducted according to the ethical standards of the Declaration of Helsinki. All individuals were recruited from the Department of Pulmonary Medicine at Tartu University Hospital in Tartu, and written informed consent was obtained from each participant prior to their recruitment.

### 2.1. Study Individuals

Patients diagnosed with a severe AE-COPD, defined as those requiring hospitalization according to the GOLD 2022 consensus document [3], and patients diagnosed as having stable COPD of all categories and degrees of severity according to GOLD 2022 [3] were included in this study. Patients with AE-COPDs that required hospitalization had their peripheral blood drawn 3–14 days after being admitted. All patients with COPD were required to have a post-bronchodilator forced expiratory volume in one second (FEV_1_)/forced vital capacity (FVC) ratio below 0.7. Patients with COPD were classified and managed according to the refined ABCD assessment tool in concordance with the GOLD 2022 consensus report [3]. Patients with AE-COPDs received bronchodilators, systemic corticosteroids, oxygen therapy, and antibiotics in agreement with the GOLD 2022 report [3]. The exclusion criteria for patients with both stable COPD and AE-COPDs included the need for mechanical ventilation, admission to an intensive care unit, unstable coronary artery disease, and the presence of active cancer. Healthy non-smokers and smokers with normal lung function were included as references. A current smoker was defined as a person who currently smoked ≥ 1 cigarette per day and a non-smoker was defined as a person who had never smoked or who had quit smoking for at least 6 months prior to the study. Smoking cessation was calculated as the time point from which the individual was completely free of tobacco use. Exposure to biomass combustion smoke was not considered in this study, as it was unlikely to influence the course of COPD pathogenesis. All participants in the study were required to be free from upper and lower respiratory tract infections (including acute bronchitis, bronchiolitis, and pneumonia), both during the study and for at least four weeks prior. However, patients with AE-COPDs were permitted to have acute respiratory infections, excluding pneumonia, as long as these infections were deemed to be causally related to the current COPD exacerbation by the treating pulmonary physician. Spirometry [17] and lung diffusing capacity [18] measurements were performed in accordance with the standards of the American Thoracic Society/European Respiratory Society, whereas multi-ethnic and Finnish reference values were used for spirometry [19] and diffusing capacity [20] parameters, respectively. The severity of airflow obstruction in patients with COPD was categorized using the post-bronchodilator FEV_1_ to GOLD grades 1–4 [3]. Multidimensional GOLD A–D stratification, where patients are categorized into four groups based on their symptoms and exacerbation history, was used to assess the changes according to the GOLD 2022 report [3]. When comparing AE-COPDs, the subgroups of the GOLD categories C1–3 and D1–3 [21] were used.

### 2.2. RNA Extraction from PBMCs

PBMCs were separated from blood using BD Vacutainer CPT tubes (Becton Dickinson, Franklin Lakes, NJ, USA), as described before [22]. RNA was extracted from PBMCs with the Trizol method according to the manufacturer’s protocol (Invitrogen, San Diego, CA, USA) and total RNA concentration was measured on a NanoDrop ND-1000 spectrophotometer (NanoDrop Technologies, Inc., Wilmington, DE, USA). RNA was stored at −80 °C until cDNA synthesis.

### 2.3. cDNA Synthesis

cDNA was synthesized via a reverse transcriptase reaction from total RNA (250 ng) using the SuperScript III enzyme (Invitrogen, Carlsbad, CA, USA) according to the manufacturers’ instructions and as previously described [22]. The conditions for reverse transcriptase reaction were as follows: incubation at 65 °C for 5 min, followed by incubation on ice for 1 min, synthesis at 50 °C for 90 min, and then inactivation at 75 °C for 15 min. cDNA was stored at −80 °C until qRT-PCR.

### 2.4. Measurement of mRNA Expression

The mRNA expression levels were detected with the TaqMan-qRT-PCR method (ABI Prism 7900HT Sequence Detection System, Applied Biosystems by Life Technologies, Waltham, MA, USA). We used Hs00153133_m1 for COX-2, Hs00167536_m1 for 5-LO, Hs00175218 for DPP4, Hs00167536_m1 for LTA_4_H, Hs00231032_m1 for HDAC2, Hs00242302_m1 for PARP-1, Hs00166575 for SOD1, Hs00609286_m1 for GSS, Hs00155249_m1 for GCLC, Hs00157694_m1 for GCLM, Hs00829989_gH for GPx, Hs00167317_m1 for GSR, and hypoxanthine phosphoribosyltransferase-1 (HPRT-1) as a house keeper (all obtained from Applied Biosystems by Life Technologies). All reactions were conducted in quadruplicate. To quantify the mRNA levels, the comparative Ct method (ΔCt value) was employed, where the amount of target transcript was normalized to the level of endogenous HPRT-1.

### 2.5. Statistical Analysis

Missing enzyme-expression-level data (12 variables) were populated by multiple imputations with random numbers within the range of the measured data, leading to 5 new data sets. The equality of baseline data among the study groups was assessed using the Kruskal–Wallis test for numeric variables and the Pearson chi-square test for nominal variables. Logistic regression analysis was used to determine whether the transcription level of enzymes was associated with COPD’s presence, its exacerbations, or smoking history. A general linear model was used to predict continuous variables based on the transcription level of enzymes. Multinomial logistic regression was performed in order to identify whether any of the genes were differentially expressed between GOLD A–D or 1–4 stratifications. Partial-least-squares discriminant analysis (PLS-DA) was used to address the grouping ability of the enzymes. All regression analyses were adjusted to age, gender, body mass index (BMI), smoking habits, the presence of current exacerbation, and lung function data. PLS-DA is a commonly employed multivariate machine learning algorithm used for classifying and interpreting metabolomics data. It is especially applicable when the number of metabolites (independent variables) is much larger than the number of data points (samples or study individuals) [23]. In this context, the advantage of PLS-DA includes handling data sets with collinear variables, along with noisy and/or missing values [23]. For collinearity diagnostics in the general linear model, as well as in the multinomial logistic regression analysis, the calculation of the variance inflation factor (VIF) was used and explanatory variables with a VIF greater than 10, indicative of significant multicollinearity, were removed. All statistical analyses were performed with the R version 4.2.1 software (R Foundation for Statistical Computing, Vienna, Austria). In addition to the basic R functions, the following packages were used: mitools, broom, dplyr, effects, ggplot2, ggstance, grDevices, mixOmics, and nnet. Regression models were in the form of Y_i_ = β_i1_ × BMI + β_i2_ × Age + β_i3_ × gender + β_i4_ × enzyme + e_i_. The described models were applied to the study groups in two settings: all participants and COPD patients only. Unless stated otherwise, the results presented are from the analyses involving all participants. All regression analyses were adjusted to participants’ age, gender, and BMI. The results obtained with logistic regression results are presented as odds ratios (ORs) and 95% confidence intervals (95% CIs). All other results are presented as regression coefficients and their 95% CIs.

## 3. Results

Altogether, 116 subjects were enrolled in this study. Different cohorts were utilized to compare the enzymes associated with systemic inflammation and GSH metabolism in PBMCs between GOLD 2022 A–D groups of COPD (Table 1, Figure 1) and GOLD grades 1–4 of airflow limitation severity (Table 2, Figure 1) [3].

### 3.1. Association of Enzymes’ Expression with Lung Function Parameters, Exacerbations, and Smoking History

Compared to stable COPD, an ongoing AE-COPD was associated with the increased expression of PARP-1 (OR = 27.3; 95% CI: 1.2–648.5, *p* = 0.042). Compared to never-smokers, the status of having ever smoked was significantly related to a lower expression of GPx (OR = 0.994, 95% CI: 0.9887–0.9997, *p* = 0.038). Compared to never-smokers and ex-smokers together, current smoking significantly decreased the expression of GSR (OR = 1.031, 95% CI: 1.001–1.062, *p* = 0.044).

The longer the time from smoking cessation became, the lower the expression levels of GPx (β = −0.02; 95% CI: −0.037–−0.004, *p* = 0.013) (Figure 2a) and GSR (β = −0.27; 95% CI: −0.505–−0.033, *p* = 0.026) (Figure 2b) were. On the contrary, higher numbers of pack-years produced significantly higher expression of GPx (β = 0.02; 95% CI: 0.002–0.038, *p* = 0.03) (Figure 2c).

Among COPD patients, lower lung function parameters, FEV_1_ % predicted (β = −5.42; 95% CI: −8.854–−1.977, *p* = 0.002) (Figure 2d), PEF % predicted (β = −3.73; 95% CI: −6.724–−0.731, *p* = 0.015), FVC % predicted (β = −6.09; 95% CI: −10.124–−2.063, *p* = 0.003), and FEV_1_ decline over years (β = 0.2; 95% CI: 0.042–0.359, *p* = 0.013) were significantly associated with higher HDAC2 levels (Figure 3a). Lower FEV_1_/FVC % predicted was connected with a higher expression of SOD1 (β = −0.13; 95% CI: −0.225–−0.036, *p* = 0.007) (Figure 3b).

Of the transfer test parameters, K_CO_ % predicted, but not K_CO_ itself, significantly decreased along with the increase in GSR expression (β = −0.642; 95% CI: −1.051–−0.232, *p* = 0.002) (Figure 3c). The decline in DL_CO_ was associated with a lower expression of LTA_4_H (β = 0.103, 95% CI: 0.001–0.205, *p* = 0.048) (Figure 3d).

### 3.2. Differences in Enzyme Expression between GOLD 1–4 Stratification

The expression of HDAC2 decreased amongst patients with COPD GOLD airflow obstruction grades 1–3, but increased amongst those with more advanced obstruction. PARP-1 was increased in GOLD grades 1–3 and GCLM rose in GOLD grades 1–2 and 4, but GCLC was decreased in GOLD grades 3–4 and DPP4 fell in GOLD grade 4 compared to non-obstructive controls, irrespective of their smoking history (Figure 4a). If non-obstructive smokers were grouped as a separate class, the expression of PARP-1, GSR, GCLM, and GCLC differed between COPD GOLD grades 1–4 and non-obstructive non-smokers, where PARP-1 was upregulated amongst milder stages or airflow obstruction but downregulated amongst patients with more severe obstruction. DPP4 and LTA_4_H levels were lower in COPD GOLD grade 4 patients than in non-obstructive non-smokers. Compared to non-obstructive non-smokers, HDAC2 expression was downregulated amongst COPD GOLD grades 1–2 and upregulated in GOLD 4. Non-obstructive non-smokers and non-obstructive smokers differed from each other only in terms of the expression of GSR and LTA_4_H (being higher and lower, respectively, in smokers) (Figure 4b).

### 3.3. Differences in Enzyme Expression between GOLD A–D Stratification

When the patients were stratified according to GOLD A–D [3], the four COPD groups had similar expression differences to each other when comparisons were made to non-obstructive individuals, irrespective of their smoking history. In particular, compared to non-obstructive individuals, PARP-1 and GCLM levels were increased amongst all GOLD groups A–D and LTA_4_H expression amongst the GOLD group A, whereas HDAC2 was decreased amongst GOLD groups A and B, DPP4 fell amongst GOLD groups B and D, and GCLC fell amongst GOLD groups A and C (Figure 4c). With A–D stratification, the results had no dependence on whether the comparator group consisted of non-obstructive non-smokers only or non-obstructive non-smokers and non-obstructive smokers together.

In patients with a AE-COPD history of ≥2 moderate exacerbations or ≥1 severe exacerbation within the last year compared to patients without AE-COPD and non-obstructive controls irrespective of their smoking history, PARP-1 expression was increased in GOLD grades 1–3 of airflow obstruction and GCLM in GOLD grades 1–2 and 4, HDAC2 expression was decreased in GOLD grades 1–3, DDP4 fell in GOLD grade 4, and GCLC fell in GOLD grades 1–2 (Figure 4d, Appendix A).

### 3.4. Grouping Ability of the Enzymes

HDAC2 showed a trend towards increase in its expression amongst patients with a history of ≥2 moderate AE-COPD incidents or ≥1 severe AE-COPD within the last year. The characteristics of the individuals involved are presented in Appendix A.

Addressing the enzymes’ ability to group COPD patients according to the severity of the disease did not reveal any significant classifying properties amongst any combination of the enzymes studied. PARP-1 and HDAC2, when grouped together, showed a trend towards association with the presence of AE-COPD.

## 4. Discussion

When analyzing the associations between the translational activity of OS- and inflammation-related enzymes with clinical COPD parameters and recommended stratification levels, our results revealed that AE-COPD was associated with PARP-1 expression. This referred to intensified DNA repair occurring in PBMCs during AE-COPD and added a valuable adjunct to the current understanding of the systemic component of AE-COPD. We and others previously showed that PARP-1 is a key player in the progression of COPD [2,11]. Currently, PARP-1 is highly expressed among GOLD classes of airflow obstruction 1–3, compared to non-obstructive controls. However, patients with high risk of AE-COPD (GOLD groups C and D) fail to distinguish A from B. In patients with milder airflow obstruction, PARP-1 expression was higher in patients with a history of AE-COPDs, compared to those with no exacerbations within the last year. This may imply that AE-COPDs already convey potentially long-lasting systemic enzymatic changes in relatively less advanced COPD. Although PARP-1 is not the only pathway to protect patients from AE-COPDs, it may serve as a potential tool with which to predict the occurrence of AE-COPDs.

We also showed the upregulation of the GSH metabolism pathway enzymes. The expression of GSR and SOD1 was increased with the decline in lung function and that of GPx and GSR decreased in connection with greater smoking history. GSH plays an important role in fighting against OS [16], where antioxidant enzymes GPx and SOD1 are the first-line defense against reactive oxygen species (ROS) and their by-products [24]. The positive association of GPx expression with having ever smoked, smoking pack-years, and smoking during the last 6 months, as well as the negative association with a longer period of time passing from smoking cessation, is connected to increased CS-induced systemic OS stress and either overwhelmed or depleted GSH responses to the OS stress in longer-term or heavier smokers. The failure or impairment of the antioxidant defense leads to more extensive oxidative damage and augmented OS in the form of chain reactions and explains the depletion of GPx in smokers [25]. In this current paper, we showed a negative interdependence between FEV_1_/FVC % and SOD1 expression. The expression of GSR was equally increased amongst COPD patients and non-obstructive smokers compared to non-obstructive non-smokers, supporting the hypothesis that the increased availability of GSH in persons with a high oxidant load may protect against lung damage [26]. In line with this, the expression of GSR declined when the time interval from smoking cessation increased and increased when K_CO_ % declined. In epithelial airway cells, GSR upregulation was shown in patients with COPD compared to non-obstructive individuals [26,27]. In PBMCs, we showed that the shifts in expression in both GSR and SOD1, along with increased airway obstruction in favor of CS, can cause extensive systemic and not only localized OS.

Our current results from PBMCs showed increased GCLM and decreased GCLC in parallel with a decline in lung function and increase in COPD severity as assessed by GOLD A–D. Bentley et al. showed a significant increase in GCLM expression in the alveolar macrophages of COPD patents compared to non-obstructive smokers [26]. Also, an increase in GCLM expression was demonstrated in the lung epithelium of patients with COPD [26,28]. GCLM has no enzymatic activity on its own but increases the catalytic efficiency of GCLC. An increase in GCLM levels would significantly impact GCL activity and GSH production in vivo, more so than an increase in GCLC levels [29].

Of the other enzymes connected with systemic inflammation, the expression of HDAC2 significantly was increased in PBMCs in parallel with a decline in lung function, suggesting the amplification of the systemic inflammatory response. Importantly, the decline in FEV_1_ % over years co-occurred with increased HDAC2 expression. PARP-1 and HDAC2 were overexpressed in various cancers and contributed to tumor progression through their roles in DNA repair, transcriptional regulation, and the maintenance of genomic integrity [30]. CS contains benzo[a]pyrene, a polycyclic aromatic hydrocarbon that has been shown to increase the activity of HDAC2 [31]. Due to this, CS might be the reason behind HDAC2 increase, rather than inflammation itself. In normal settings, increased HDAC2 level should decrease the acetylation of PARP-1, but amongst COPD patients, the increase in PARP-1 can be so significant that the balance between normal HDAC2 and PARP-1 is markedly shifted. The increase in the systemic HDAC2 expression along with the decline in lung function, however, was stronger in COPD patients than among non-obstructive smokers and non-smokers, meaning that the tendency for HDAC2 to increase in expression may be more strongly related to lung function decline than to smoking alone.

Our analysis revealed that DPP4 expression was reduced in COPD patients who exhibited more severe airflow obstruction and had a history of AE-COPD. Relying on the former insight, the significant association with the shifts in DPP4 expression and the highest stages of COPD potentially suggests a diabetic or cardiovascular origin for this association [32].

In contrast to 5-LO and COX2, a downstream leukotriene metabolizing enzyme LTA_4_H had significant associations with the decline in DL_CO_, a parameter that depends not solely on lung structure and volume, but which is also influenced by lung perfusion and pulmonary hypertension [33]. LTA_4_H is present in many cells that lack significant 5-LO activity and can also be transported to areas in need by blood cells; due to that, LTA_4_H can be more sensitive to changes in the severity of the disease in PBMCs [9].

The limitations of this study include small cohort sizes in certain subgroups, which could reduce the magnitude of the differences found. As a mitigation measure, cohorts were pooled for multinominal logistic regression analysis. Secondly, the peripheral blood samples were obtained 3–14 days after hospital admission and this only concerned patients hospitalized with AE-COPD. This was also a single-center study and the involvement of a wider population might have increased the significance of our results. As there were no preliminary data, no sample size calculations were performed before this study. In addition, future studies using whole-genome microarray analysis could provide new knowledge of what other genes may be involved in AE-COPD and COPD progression. Further, protein-expression-level studies could offer valuable insights into identifying biomarkers for the disease. To counterbalance, the strengths of our current research include the large number of enzymes measured, the prospective design, and adequacy of the statistics used.

The results are presented in a schematic format in Figure 5.

## 5. Conclusions

Our results indicate that GSH metabolism and pro-inflammatory gene expression may contribute to the damage to the lungs that characterizes the chronic nature of COPD. They also highlight the importance of a network of genes in the lungs’ response to OS and systemic inflammation. Importantly, not all genes see their expression increase or decrease as the airflow limitation progresses, raising important questions for further research as to what might changes might associated with lung function decline or the development of COPD. Our results also support the finding that systemic inflammation and OS do not necessarily correlate with the GOLD A–D COPD classification. The further integration of association and expression studies to determine the nature of the biological relationships may lead to a better understanding of the disease and, from there, to developments that may potential support the control of the disease.

In summary, our findings suggest that GSH metabolism and pro-inflammatory gene expression contribute to lung damage in COPD, while also raising questions for further research about the relationship between gene expression changes, COPD progression, and AE-COPD.

## Figures and Tables

**Figure 1 antioxidants-13-00953-f001:**
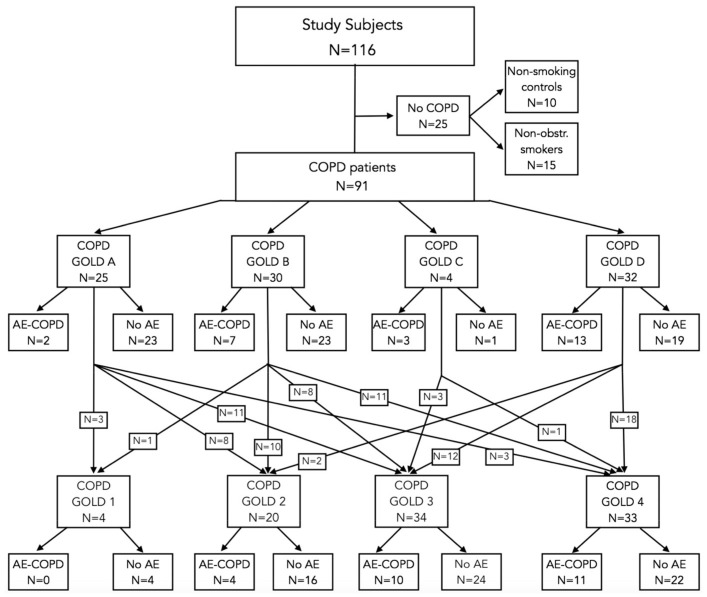
Flow diagram of patient distribution in terms of chronic obstructive pulmonary disease (COPD) categories by degrees of severity and airflow limitation according to the Global Initiative for Chronic Obstructive Lung Disease (GOLD) 2022 [3], also showing non-obstructive smokers and normal control individuals. Distribution of COPD patients according to multidimensional GOLD A–D stratification. GOLD grades 1–4 of airflow limitation severity are shown by presence or absence of acute exacerbations of COPD (AE-COPDs).

**Figure 2 antioxidants-13-00953-f002:**
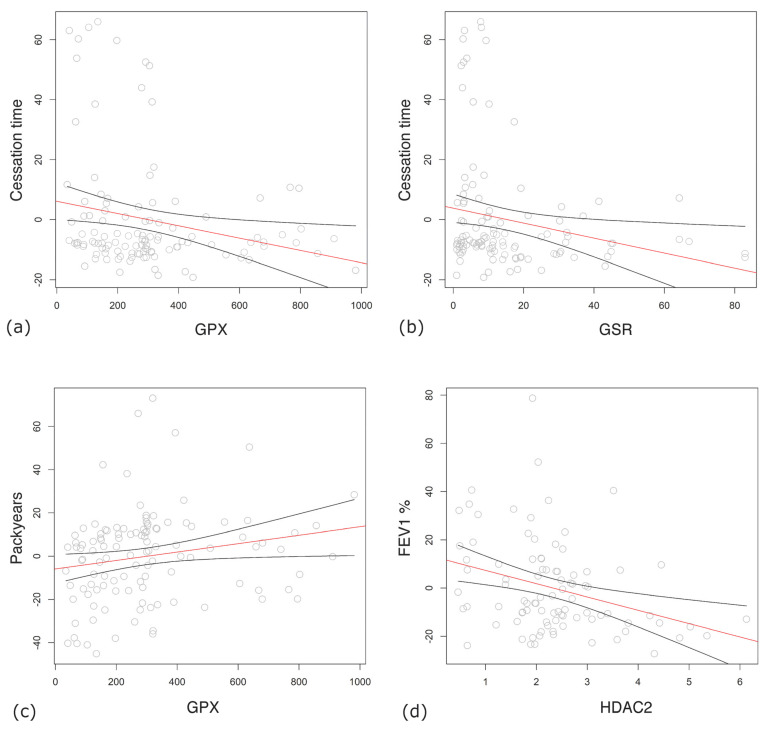
Transcription levels of enzymes associated with lung function parameters, as analyzed by general linear model. Residual errors after correction to age, gender, and body mass index (BMI) vs. mRNA expression levels of enzymes and estimated regression line with 95% confidence intervals are shown. (**a**) Time from smoking cessation (months) vs. glutathione peroxidase (GPx). (**b**) Time from smoking cessation (months) vs. glutathione reductase (GSR). (**c**) Number of smoking pack-years vs. GPx. (**d**) Forced expiratory volume in one second (FEV_1_) vs. histone deacetylase 2 (HDAC2). Regression lines are shown in red with their lower and upper 95% confidence limits as black curves.

**Figure 3 antioxidants-13-00953-f003:**
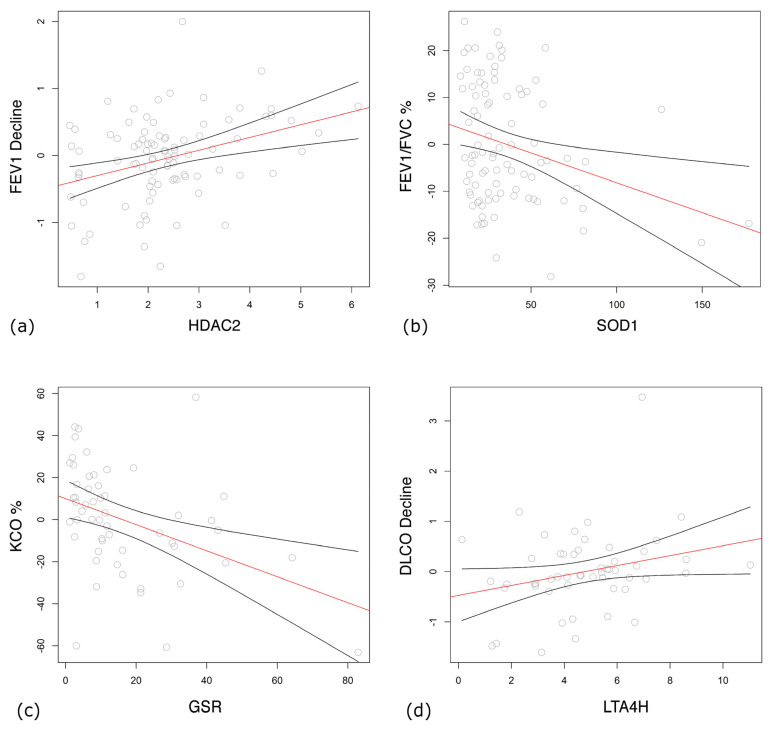
Transcription levels of enzymes associated with lung function parameters were analyzed by general linear model. Residual errors after correction to age, gender and body mass index (BMI) vs. mRNA expression levels of enzymes and estimated regression line with 95% confidence intervals are shown. (**a**) Forced expiratory volume in one second (FEV_1_) decline over years vs. histone deacetylase 2 (HDAC2). (**b**) FEV_1_/forced vital capacity (FVC) % vs. superoxide dismutase (SOD1). (**c**) Carbon monoxide transfer coefficient (K_CO_) % vs. glutathione reductase (GSR). (**d**) Diffusing capacity of carbon monoxide (DL_CO_) decline vs. leukotriene A4 hydrolase (LTA_4_H). Regression lines are shown in red with their lower and upper 95% confidence limits as black curves.

**Figure 4 antioxidants-13-00953-f004:**
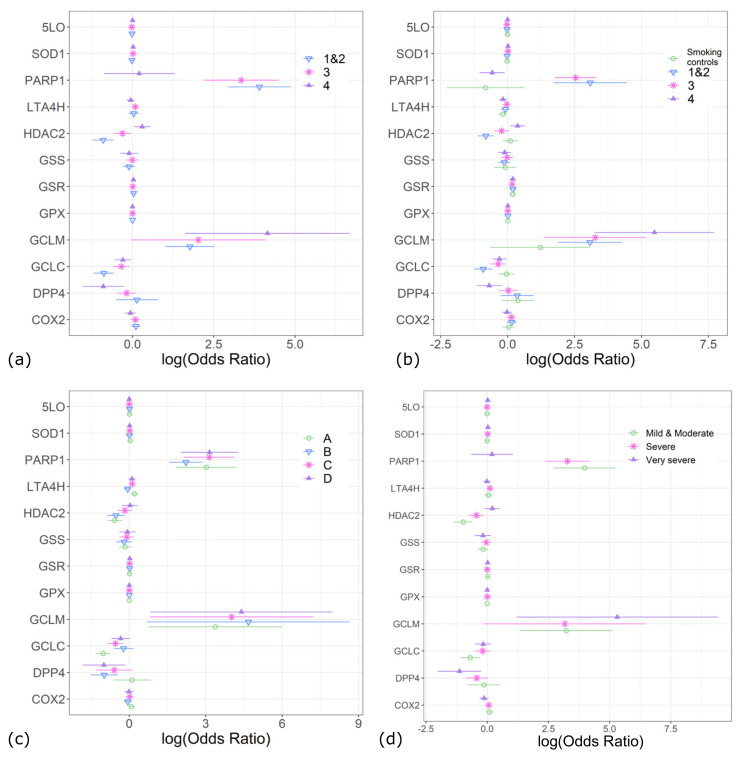
Differences in levels of expression of mRNA of enzymes involved in glutathione (GSH) metabolism and inflammation, as analyzed by multinominal logistic regression. Natural logarithms of odds ratios vs. comparators with 95% confidence intervals for each enzyme are shown. Data are adjusted to age, gender, and body mass index (BMI). (**a**) Patients with chronic obstructive pulmonary disease (COPD), assessed by airflow limitation severity according to Global Initiative for Chronic Obstructive Lung Disease (GOLD) classification 1–4 [3], compared to all non-obstructive controls irrespective of smoking history. (**b**) Non-obstructive smokers and patients with COPD, assessed by airflow limitation severity according to GOLD classification 1–4 [3], compared to non-obstructive non-smokers. (**c**) Patients with COPD according to GOLD classes A–D [3] compared to pooled non-obstructive controls irrespective of their smoking history. (**d**) Patients who have experienced at least two moderate exacerbations or at least one severe exacerbation in past year compared to all individuals who have not experienced an exacerbation (non-obstructive non-smokers, non-obstructive smokers, and patients with COPD as assessed by airflow limitation severity according to GOLD classification 1–4 [3] without COPD exacerbation).

**Figure 5 antioxidants-13-00953-f005:**
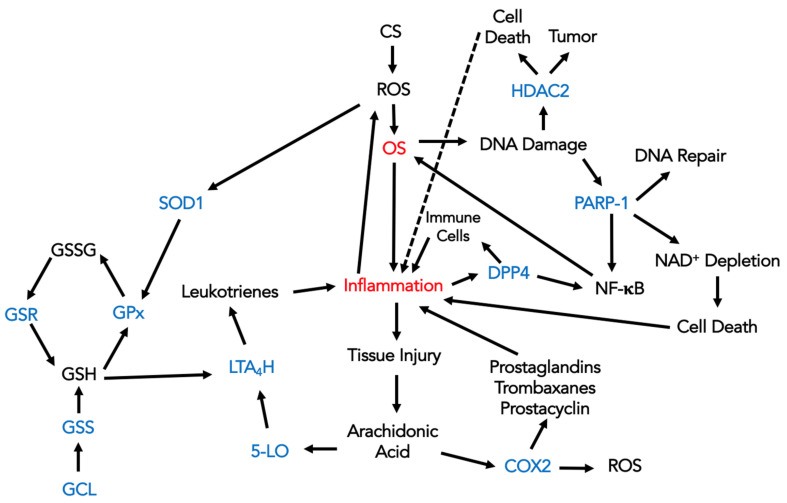
A schematic diagram explaining how the enzymes involved in inflammation and GSH pathways, whose mRNA expression was currently studied (marked with blue), are linked to OS and inflammation caused by ROS from environmental agents, primarily from CS, as well as from cellular responses. 5-LO—5-lipoxygenase; CS—cigarette smoke; COX2—cyclooxygenase-2; DPP4—dipeptidyl peptidase 4; DNA—deoxyribonucleic acid; GCL—glutamyl–cysteine ligase; GPx—glutathione peroxidase; GSH—glutathione; GSR—glutathione reductase; GSS—glutathione synthetase; GSSG—glutathione disulfide; HDAC2—histone deacetylase; LTA4H—leukotriene A4 hydrolase; mRNA—messenger ribonucleic acid; NAD^+^—nicotinamide adenine dinucleotide; NF-κB—nuclear factor kappa-light-chain-enhancer of activated B cells; OS—oxidative stress; PARP-1—poly(ADP-ribose) polymerase-1; ROS—reactive oxygen species; SOD1—superoxide dismutase 1.

**Table 1 antioxidants-13-00953-t001:** Characteristics of the individuals, including the measurement of the mRNA expression of enzymes associated with systemic inflammation and glutathione (GSH) metabolism in peripheral blood mononuclear cells. Patients were diagnosed with the exacerbation of chronic obstructive pulmonary disease (COPD) and patients were diagnosed as having stable COPD of all categories and degrees of severity of airflow obstruction in accordance with the Global Initiative for COPD (GOLD) consensus document 2022 [3].

Characteristics	Non-Smoking Controls (*n* = 10)	Non-Obstructive Smokers (*n* = 15)	COPD				
			GOLD A (*n* = 25)	GOLD B (*n* = 30)	GOLD C (*n* = 4)	GOLD D (*n* = 32)	*p*-Values **
Age	64.0 ± 3.4	61.1 ± 2.9	69.0 ± 2.5	67.1 ± 2.0	80.0 ± 2.7	71.7 ± 1.6	0.008
Male	7 (70%)	9 (60%)	21 (84%)	29 (97%)	4 (100%)	31 (97%)	0.004
BMI	25.7 ± 1.6	27.0 ± 1.6	25.8 ± 1.0	23.4 ± 0.8	22.1 ± 0.6	24.0 ± 0.9	0.138
Smoking (pack-years)	-	35.7 ± 2.6	40.4 ± 3.8	39.5 ± 3.6	45.0 ± 5.0	43.5 ± 4.4	0.799
Current smoker	-	10 (67%)	17 (68%)	16 (53%)	0 (0%)	13 (41%)	0.038
Smoking cessation amongst ex-smokers (years ago)	-	6.8 ± 3.5	12.1 ± 3.4	9.0 ± 2.5	7.6 ± 3.1	7.9 ± 1.9	0.061
FEV_1_ % predicted	97.6 ± 4.5	82 ± 4.5	53.6 ± 4.7	41.1 ± 3.4	34.8 ± 3.0	30.0 ± 1.8	<0.001
Absolute decline in FEV_1_ % over years (%/year) *	0.1 ± 0.1	0.6 ± 0.1	1.4 ± 0.4	1.6 ± 0.2	1.2 ± 0.1	1.6 ± 0.1	<0.001
Current exacerbation	-	-	2 (8%)	7 (23%)	3 (75%)	13 (41%)	0.006
PEF % predicted	99.5 ± 6.8	80.6 ± 4.1	43.7 ± 4.3	35.1 ± 2.9	27.8 ± 5.5	27.3 ± 1.6	<0.001
FVC % predicted	96.8 ± 4.7	79.9 ± 4.7	70.8 ± 4.8	59.2 ± 3.8	53.8 ± 7.8	46.7 ± 2.7	<0.001
FEV_1_/FVC %	81.9 ± 1.2	82.3 ± 1.6	62.7 ± 1.9	60.9 ± 1.6	55.8 ± 2.6	52.5 ± 1.5	<0.001
K_CO_	1.2 ± 0.1	1.2 ± 0.1	1.0 ± 0.1	0.8 ± 0.1	0.6 ± 0.0	0.6 ± 0.1	0.020
K_CO_ %	93.4 ± 4.8	80.3 ± 8.9	76.4 ± 7.5	56.6 ± 5.6	48.8 ± 0.0	40.5 ± 9.5	0.005
DL_CO_	6.3 ± 0.6	5.6 ± 0.8	4.8 ± 0.6	3.5 ± 0.4	2.8 ± 0.0	2.8 ± 0.6	0.014
DL_CO_ %	77.3 ± 9.0	61.6 ± 7.4	54.4 ± 5.3	38.7 ± 3.8	30 ± 0.0	31.3 ± 6.1	0.002
TLC	5.3 ± 0.3	4.82 ± 0.3	4.8 ± 0.3	4.6 ± 0.2	4.8 ± 0.0	4.6 ± 0.3	0.716
TLC %	84.9 ± 5.8	79.2 ± 3.4	73.9 ± 2.1	68.0 ±2.5	63.4 ± 0.0	67.4 ± 3.2	0.021

Data are presented as mean ± SEM or n (%). * Annual change in FEV_1_% predicted after the age of 25, assuming FEV_1_% was 100% at the age of 25 years. BMI—body mass index; DL_CO_—diffusing capacity of the lungs for carbon monoxide; FEV_1_—forced expiratory volume in one second; FVC—forced vital capacity; K_CO_—carbon monoxide transfer coefficient; PEF—peak expiratory flow; TLC—total lung capacity. ** To assess the equality of data among the study groups, the Kruskal–Wallis test was used on numeric variables, while Pearson’s chi-square test was applied to nominal variables. In the case of smoking-related variables, non-smoking controls were omitted from the analysis. In the case of current acute exacerbation of COPD, non-smoking control individuals and non-obstructive controls were omitted from the analysis.

**Table 2 antioxidants-13-00953-t002:** Characteristics of the individuals included in the measurement of the mRNA expression of enzymes associated in systemic inflammation and glutathione (GSH) metabolism in peripheral blood mononuclear cells. Patients diagnosed as having stable chronic obstructive pulmonary disease (COPD) were grouped by airflow limitation severity grades 1–4 according to the Global Initiative for COPD consensus document 2022 [3].

Characteristics	Non-Smoking Controls (*n* = 10)	Non-Obstructive Smokers (*n* = 15)	Severity of Airflow Limitation				
			GOLD 1 (*n* = 4)	GOLD 2 (*n* = 20)	GOLD 3 (*n* = 34)	GOLD 4 (*n* = 33)	*p*-Values **
Age	64.0 ± 3.4	61.1 ± 2.9	76.8 ± 3.0	67.9 ± 2.1	70.1 ± 2.0	70.0 ± 2.0	0.029
Male	7 (70%)	9 (60%)	4 (100%)	19 (95%)	32 (94%)	30 (91%)	0.010
BMI	25.7 ± 1.6	27.0 ± 1.6	23.4 ± 1.7	23.8 ± 1.1	25.5 ± 0.7	23.2 ± 0.9	0.093
Smoking (pack-years)	-	35.7 ± 2.6	56.3 ± 10.3	41.7 ± 3.1	38.2 ± 3.5	42.8 ± 4.4	0.236
Current Smoker	-	10 (67%)	2 (50%)	13 (65%)	15 (44%)	16 (48%)	0.471
Smoking cessation amongst ex-smokers (years ago)	-	6.8 ± 3.5	5.0 ± 1.0	8.0 ± 3.8	13.1 ± 2.2	10.4 ± 2.5	0.093
Current exacerbation	-	-	0 (0%)	4 (20%)	10 (29%)	11 (33%)	0.439
PEF % predicted	99.5 ± 6.8	80.6 ± 4.1	71.0 ± 14.4	51.4 ± 3.6	32.6 ± 1.5	22.4 ± 0.9	<0.001
FEV_1_ % predicted	97.6 ± 4.5	82 ± 4.5	94.5 ± 9.0	60.7 ± 2.1	38.7 ± 1.1	23.4 ± 0.8	<0.001
Absolute decline in FEV_1_ % over years (%/year) *	0.1 ± 0.1	0.6 ± 0.1	0.1 ± 0.2	1.0 ± 0.1	1.7 ± 0.3	1.9 ± 0.1	<0.001
FVC % predicted	96.8 ± 4.7	79.9 ± 4.7	110.3 ± 7.7	80.0 ± 2.6	57.7 ± 1.7	38.0 ± 1.4	<0.001
FEV_1_/FVC %	81.9 ± 1.2	82.3 ± 1.6	64.7 ± 2.5	60.7 ± 1.3	54.6 ± 1.5	51.8 ± 2.2	<0.001

Data are presented as mean ± SEM or n (%). * Annual change in FEV_1_ % predicted after the age of 25, assuming FEV_1_ % was 100% at the age of 25 years. BMI—body mass index; FEV_1_—forced expiratory volume in one second; FVC—forced vital capacity; PEF—peak expiratory flow. ** To assess the equality of data among the study groups, the Kruskal–Wallis test was used for numeric variables, while Pearson’s chi-square test was applied to nominal variables. In the case of smoking-related variables, non-smoking controls were omitted from the analysis. In the case of current acute exacerbation of COPD, non-smoking control individuals and non-obstructive controls were omitted from the analysis.

## Data Availability

Data are available upon request by mailing to the corresponding author.

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
