# Peer review of "Pattern of Expression of Genes Involved in Systemic Inflammation and Glutathione Metabolism Reveals Exacerbation of COPD"

_antioxidants, 2024, doi:10.3390/antiox13080953_

Round 1
Reviewer 1 Report
In the Article - “A pattern of expression of genes involved in systemic inflammation and glutathione metabolism reveals exacerbation of COPD” – the authors used a logistic regression model to determine whether the transcription level of 12 enzymes is associated with COPD presence, exacerbations, or smoking history. Non-smokers were included as a control. Correlation and interdependence are presented.
1. It would be more informational to readers to see where these transcripts are located within the pathway(s). Including a figure in your paper is beneficial.
2. Can you confirm if the gene expression levels in your results are comparable with those of protein expression levels (in blood)?
3. Ideally, this study should be done by whole-genome microarray analysis. At a minimum, this should be discussed.
4. All studied transcripts can be increased rapidly right after smocking (~5 min) and decreased afterward. Was the time after/before smocking unified between the COPD group participants?
5. Please discuss/compare the parallel increase of PARP-1 and HDAC2 in lung cancer studies and in your study.
n/a
Author Response
We thank the reviewer for interest towards our manuscript and the valuable time that has been taken in thorough analysis of the manuscript and making useful recommendations for improvement of our manuscript “A pattern of expression of genes involved in systemic inflammation and glutathione metabolism reveals exacerbation of COPD”.
We have provided point-by point answers to the comments and revised our manuscript in the light of all suggestions and requirements. Modifications in the manuscript have been marked in blue. English language revisions are marked in red.
In the Article - “A pattern of expression of genes involved in systemic inflammation and glutathione metabolism reveals exacerbation of COPD” – the authors used a logistic regression model to determine whether the transcription level of 12 enzymes is associated with COPD presence, exacerbations, or smoking history. Non-smokers were included as a control. Correlation and interdependence are presented.
- It would be more informational to readers to see where these transcripts are located within the pathway(s). Including a figure in your paper is beneficial.
Answer: Thank you for this valuable annotation. A sentence “Schematic format of the results is illustrated in figure 5.“ and a new figure (Figure 5) with its figure text has now been included in the article to illustrate the relationship among all 12 enzymes.
Figure 5. A schematic diagram explaining how the enzymes involved in inflammation and GSH pathway, whose mRNA expression was currently studied (marked with blue), are linked to OS and inflammation caused by ROS from environmental agents, primarily from CS, as well as from cellular responses.
5-LO – 5-lipoxygenase, CS – cigarette smoke, COX2 – cyclooxygenase-2, DPP4 – dipeptidyl peptidase 4, DNA – deoxyribonucleic acid, GCL – glutamyl-cysteine ligase, GPx – glutathione peroxidase, GSH – glutathione, GSR – glutathione reductase, GSS – glutathione synthetase, GSSG – glutathione disulfide, HDAC2 – histone deacetylase, LTA4H – leukotriene A4 hydrolase, mRNA – messenger ribonucleic acid, NAD+ – nicotinamide adenine dinucleotide, NF-κB – nuclear factor kappa-light-chain-enhancer of activated B cells, OS – oxidative stress, PARP-1 – poly(ADP-ribose) polymerase-1, ROS – reactive oxygen species, SOD1 – superoxide dismutase 1.
Please refer to page 12, lines 392-405 in the revised manuscript.
- Can you confirm if the gene expression levels in your results are comparable with those of protein expression levels (in blood)?
Answer: That is a great question. The correlation between mRNA expression and protein expression has been described as relatively poor, but despite the poor correlation, differential mRNA expression is still considered biologically significant. Changes in mRNA levels can indicate potential changes in protein function, especially when mRNA is significantly differentially expressed under varying conditions. In our study we did not confirm our finds via protein expression levels. This is a step for our future studies.
A sentence has also been added to limitations. “In addition, future studies using whole-genome microarray analysis could give new knowledge of what other genes may be involved in AE-COPD and COPD progression, as well as protein expression level studies could offer valuable insights into identifying biomarkers for the disease.” Please look at the page 11, lines 386-388 and page 12, lines 389-390.
- Ideally, this study should be done by whole-genome microarray analysis. At a minimum, this should be discussed.
Answer: That is a great observation. Indeed, whole-genome microarray analysis is a powerful technique to study genomic variations across the entire genome. However, microarray analysis provides relative quantitation of enzyme expression, and the results usually need to be validated, for example, by qRT-PCR. Whole-genome microarray analysis indicates the expression levels of the genes across the entire genome simultaneously. qRT-PCR quantifies specific RNA transcripts by amplifying them in real-time using PCR allowing determination of the initial quantity of the target RNA. qRT-PCR is also more sensitive to detect low-abundance transcripts. It is considered the gold standard for confirming gene expressions due to its high specificity and reproducibility. The aim of our current study was to determine the changes amongst these 12 specific enzymes mRNA changes, but we agree that whilst performing whole-genome microarray analysis, new and interesting enzymes may come up.
A sentence was added to the limitations: “In addition, future studies using whole-genome microarray analysis could give new knowledge of what other genes may be involved in AE-COPD and COPD progression, as well as protein expression level studies could offer valuable insights into identifying biomarkers for the disease.” Please refer to page 11, lines 386-388 and page 12, lines 389-390.
- All studied transcripts can be increased rapidly right after smocking (~5 min) and decreased afterward. Was the time after/before smocking unified between the COPD group participants?
Answer: We thank this reviewer for picking up this issue. Reasonably, this issue concerned current smokers. Peripheral blood was drawn after spirometry was performed and after patient consent form was read and signed, giving us at least an hour window, where no subject that took part in our study had the chance to smoke.
- Please discuss/compare the parallel increase of PARP-1 and HDAC2 in lung cancer studies and in your study.
Answer: This is a great question again. PARP-1 and HDAC2 are overexpressed in various cancers and contribute to tumor progression through their roles in DNA repair, transcriptional regulation, and maintenance of the genomic integrity. HDAC2 interacts with PARP-1, modulating its activity. This interaction is important for the regulation of NF-κB-dependent transcription, which is involved in inflammatory responses. By reducing the acetylation of PARP-1, HDAC2 facilitates maintaining a balance in inflammatory signaling pathways. HDAC2 normally suppresses the expression of pro-inflammatory genes by deacetylating core histones, however, it has been shown that CS increases the activity of HDAC2. Due to that, CS might be the reason behind the HDAC2 increase, rather than inflammation itself. In normal setting, increased HDAC2 level should decrease the acetylation of PARP-1, but amongst COPD patients, the increase in PARP-1 could be so significant, that the balance between normal HDAC2 and PARP-1 is markedly shifted.
We have added two sentences into our discussion. “PARP-1 and HDAC2 are overexpressed in various cancers and contribute to tumor progression through their roles in DNA repair, transcriptional regulation, and maintenance of genomic integrity [30].” and “Due to that, CS might be the reason behind HDAC2 increase, rather than inflammation itself. In normal setting, increased HDAC2 level should decrease the acetylation of PARP-1, but amongst COPD patients, the increase in PARP-1 could be so significant, that the balance between normal HDAC2 and PARP-1 is markedly shifted.” In association, a new citation (30) was added “Kruglov, O., Wu, X., Hwang, S. T., Akilov, O. E., The synergistic proapoptotic effect of PARP-1 and HDAC inhibition in cutaneous T-cell lymphoma is mediated via Blimp-1. Blood Adv, 2020. 4(19): p. 4788-4797”
Please see page 11, lines 357-360 and 361-365; page 21, lines 570-571.
Reviewer 2 Report
The main question addressed by the research is that GSH metabolism and proinflammatory gene expression may contribute to damage to the lungs that characterizes the chronic nature of chronic obstructive pulmonary disease (COPD). They profiled mRNA expression of enzymes connected to systemic inflammation and GSH metabolism in peripheral blood mononuclear cells (PBMC) from clinical patients which is original. The article demonstrated the associations of COPD to the expression of enzymes linked to inflammation and regulation of GSH metabolism.
The data were of high quality and clearly illustrated. Controls used in this work are proper. The conclusions are consistent with the evidence and arguments presented. The references are appropriate. Overall, this work is suitable for publication after addressing a few minor concerns.
Comments and questions:
1. Although the references of GOLD grades 1-4 and GOLD A-D have been listed, it’s better to briefly introduce what they are to help understand the data in this paper.
2. Supply the second title for Section 3 ‘Results’.
3. The resolution of Figure 2 and Figure 3 is low.
4. Emphasis why chooses the four enzymes (HDAC2, SOD1, GSR, and LTA4H) in figure 2 and figure 3 form more than 10 enzymes that mentioned in the introduction?
5. Draw a brief conclusion after summarizing the data in the table and figures.
In summary, I recommended minor revisions to this article before its publication in Antioxidants.
Comments and questions:
1. Although the references of GOLD grades 1-4 and GOLD A-D have been listed, it’s better to briefly introduce what they are to help understand the data in this paper.
2. Supply the second title for Section 3 ‘Results’.
3. The resolution of Figure 2 and Figure 3 is low.
4. Emphasis why chooses the four enzymes (HDAC2, SOD1, GSR, and LTA4H) in figure 2 and figure 3 form more than 10 enzymes that mentioned in the introduction?
5. Draw a brief conclusion after summarizing the data in the table and figures.
In summary, I recommended minor revisions to this article before its publication in Antioxidants.
Author Response
We thank the reviewer for interest and the valuable time that has been taken in thorough analysis of the manuscript and making useful recommendations for improvement of our manuscript “A pattern of expression of genes involved in systemic inflammation and glutathione metabolism reveals exacerbation of COPD”.
We have provided point-by point answers to the comments and revised our manuscript in the light of all suggestions and requirements. Modifications in the manuscript have been marked in blue. English language revisions are marked in red.
The author hypothesized that GSH metabolism and proinflammatory gene expression may contribute to damage to the lungs that characterize the chronic nature of chronic obstructive pulmonary disease (COPD). To test the hypothesis of serving as systemic biomarkers of COPD, they profiled mRNA expression of enzymes connected to systemic inflammation and GSH metabolism in peripheral blood mononuclear cells (PBMC) from patients. The results demonstrate associations of COPD to the expression of enzymes linked to inflammation and regulation of GSH metabolism.
The data were of high quality and clearly illustrated. Controls used in this work are proper. Overall, this work is suitable for publication after addressing a few minor concerns.
- Although the references of GOLD grades 1-4 and GOLD A-D have been listed, it’s better to briefly introduce what they are to help understand the data in this paper.
Answer: Thank you for this valuable annotation. Indeed, the GOLD grades 1-4 refer only to the severity of the airflow obstruction in the airways (though before 2011, it was expected to classify the whole COPD). After realizing that COPD represents much more than just airflow obstruction seen with spirometry, later on, multidimensional GOLD A–D stratification was used to assess the changes, based on their symptoms and exacerbation history. And since 2017 onward, moreover, the spirometric lung function was not included anymore as an input to classify COPD. There is a brief explanation is in the “study individuals” part: “The severity of airflow obstruction in patients with COPD was categorized according to the post-bronchodilator FEV1 to GOLD grades 1-4 [3]. Multidimensional GOLD A–D stratification was used to assess the changes according to the GOLD 2022 report [3].“
Longer explanation was added to reflect the classification in more detail: “Multidimensional GOLD A–D stratification, where the patients were categorized into four groups based on their symptoms and exacerbation history, was used to assess the changes according to the GOLD 2022 report [3].“ Please refer to page 3, lines 102-104.
- Supply the second title for Section 3 ‘Results’.
Answer: Second titles were added to “Results”:
3.1 Association of enzymes’ expression with lung function parameters, exacerbations, and smoking history
3.2 Differences in enzyme expression between GOLD 1-4 stratification
3.3 Differences in enzyme expression between GOLD A-D stratification
3.4 Grouping ability of the enzymes
Please refer to page 6, line 214-215; page 8, line 252; page 9, line 282; and page 10, line 300.
- The resolution of Figure 2 and Figure 3 is low.
Answer: Thank you for pointing out this issue. The figures 2 and 3 have been changed. Now, all figures are above the required resolution (a minimum of 1000 pixels in width/height or a resolution of 300 dpi or higher).
- Emphasis why chooses the four enzymes (HDAC2, SOD1, GSR, and LTA4H) in figure 2 and figure 3 form more than 10 enzymes that mentioned in the introduction?
Answer: That is a great observation. The most illustrative enzymes were selected to enhance visual representation. If all the enzymes had been added, there would have been an overwhelming number of images. All significant results are presented in the article text.
- Draw a brief conclusion after summarizing the data in the table and figures.
Answer: A brief conclusion has been added to the end of the article: “In summary, our findings suggest that GSH metabolism and proinflammatory gene expression contribute to lung damage in COPD, while raising questions for further research about the relationship between gene expression changes and COPD progression, as well as AE-COPD.”
Please look at the page 12, lines 417-420.